# A New Role for Epidurography: A Simple Method for Assessing the Adequacy of Decompression during Percutaneous Plasma Disc Decompression

**DOI:** 10.3390/jcm11237144

**Published:** 2022-12-01

**Authors:** Ho Young Gil, Wonseok Seo, Gyu Bin Choi, Eunji Ha, Taekwang Kim, Jungyul Ryu, Jae Hyung Kim, Jong Bum Choi

**Affiliations:** 1Department of Anesthesiology and Pain Medicine, Ajou University School of Medicine, Suwon 16499, Republic of Korea; 2Department of Anesthesiology and Pain Medicine, Dongtan Sacred Heart Hospital, Hallym University School of Medicine, Hwaseong 18450, Republic of Korea

**Keywords:** decompression, discogenic low back pain, epidurography, intervertebral disc displacement, low back pain, lumbar radicular pain, nucleoplasty, outcome, percutaneous plasma disc decompression

## Abstract

Percutaneous plasma disc decompression (PPDD) is a minimally invasive treatment for discogenic low back pain and herniated disc-related symptoms. However, there are no known outcome predictive variables during the procedure. The purpose of this study was to evaluate and validate epidurography as an intra-procedure outcome predictor. We retrospectively enrolled 60 consecutive patients who did not respond to conventional treatments. In the next stage of treatment, PPDD was performed, and the epidurography was conducted before and after the PPDD. We analyzed the relationship between epidurographic improvement and the success rate. The Numerical Rating Scale and the Oswestry Disability Index were used to assess pain and functional capacity, respectively, before the procedure and 1 month after the procedure. The pain reduction and the success rate in the epidurographic improvement group were significantly higher than in the epidurographic non-improvement group. Both the Numerical Rating Scale and the Oswestry Disability Index scores were significantly reduced in both groups, but there was no significant difference in Oswestry Disability Index scores. This study’s results showed that PPDD is an effective treatment method. We also suggested that epidurography may be a potential outcome predictor for ensuring successful outcomes and determining the endpoint of the procedure.

## 1. Introduction

Low back pain is very common and affects 80% of individuals at some point in their lives [1,2]. According to one study, about 40% of chronic low back pain is caused by discogenic back pain, and most discogenic back pain is caused by disc prolapse or degenerative disc disease. Although these are common findings in asymptomatic patients, provocative discography has been used to distinguish between a painful disc and a nonpainful disc [3].

Management of chronic discogenic low back pain includes noninvasive conservative treatments, such as medications, physical therapy, behavior management, psychotherapy, and invasive surgical approaches. In a recent meta-analysis, it was argued that surgical treatment is not superior to non-surgical treatment as a treatment for chronic low back pain [4]. In addition, the range of indications for surgery is extremely small; it includes the paralysis of functionally important muscles and cauda equina syndrome, but not most patients with disc herniations.

Percutaneous plasma disc decompression (PPDD) was approved in the USA by the Food and Drug Administration (FDA) as a minimally invasive technique for discogenic back pain in 2000 [5]. It is possible to reduce intradiscal pressures and disc volume through ablation and coagulation (Coblation^®^ technique) using bipolar radiofrequency energy to remove disc material [6]. PPDD is a minimally invasive procedure with few complications, and its effects on long-term pain reduction and functional improvement have been previously demonstrated [7].

To evaluate the success of the procedure, standardized pain measuring tools—e.g., a visual analog scale (VAS); numerical rating scale (NRS); and standardized assessments of functional capacity in spinal mobility deficit caused by back pain, such as the Oswestry Disability Index (ODI)—were used. In addition, several imaging modalities, such as plain radiography, computed tomography (CT), and magnetic resonance imaging (MRI), were used to evaluate the success of the procedure. However, these modalities are expensive, do not offer real-time assessment during the intra-procedure period, and may take longer to show changes in disc volume. Moreover, their results do not always coincide with the patients’ symptoms [8].

Although there are studies on variables that can predict and evaluate the success of PPDD before and after the procedure, there is no study on intra-procedure outcome predictive variables [6,9]. In order to increase the PPDD success rate, the physician’s technique to accurately place the needle on the target disc is required. However, since the process of decompression is made by the manufactured guideline, outcome predictive variables that enable real-time assessment during the procedure are still required [10,11,12].

When disc protrusion occurs in the epidural space, the contrast medium does not spread well on the epidurography. Consequently, epidurography was used during percutaneous epidural neuroplasty and spinal surgery to confirm the success of the procedure [13,14].

We speculated that PPDD would reduce the disc volume, ensuring the epidural space and resulting in the improvement of epidurography. Thus, the purpose of this study was to evaluate and validate epidurography as an intra-procedure outcome predictor by correlating the change in epidurography with the change in the pain and functional score after PPDD.

## 2. Materials and Methods

### 2.1. Patients

This retrospective observational study was approved by the Institutional Review Board of Ajou University Hospital of Korea (IRB No. AJIRB-MED-MDB-19-409) in November 2019. The requirement for informed consent was waived because of the retrospective case-control nature of the study.

From January 2017 to December 2019, we retrospectively enrolled 60 consecutive patients with low back pain, with or without leg radicular pain (NRS ≥ 4), who did not respond to physical therapy, medications, and epidural steroid injections [15]. As the next step of treatment, PPDD was performed. The diagnosis was based on the patient’s symptoms, neurological examination, and imaging studies. Inclusion criteria were: (1) age between 20 and 80 years old, (2) low back pain with or without leg radicular pain, (3) unresponsiveness to conservative therapy for more than three months, (4) MRI evidence of contained disc protrusion, (5) preservation of a disc height of ≥50%, (6) accurate identification of the symptomatic disc level prior to the procedure, (7) discography only if the physician is not sure about the treating level of the lumbar disc, and (8) PPDD with insertion of a wire-type epidural catheter. Exclusion criteria were: (1) disc height < 50%, (2) evidence of sequestration disc, (3) moderate/severe spinal stenosis, (4) previously operated segments, (5) spinal instability, (6) loss of follow-up, (7) inability to evaluate the outcome of PPDD because of other severe diseases, such as cancer, infection, and fracture, and (8) incomplete medical records. Patients were regularly followed up until 1 month after PPDD.

### 2.2. Percutaneous Plasma Disc Decompression

PPDD was performed on an outpatient basis by a pain physician with more than 10 years of experience in the field. (Figure 1). A prophylactic dose of 1 g cefazolin was administered intravenously 1 h prior to the procedure. If necessary, discography was performed prior to PPDD. The patient was placed in a prone position under sterile conditions. The involved disc space was localized under fluoroscopic guidance, and the soft tissues were infiltrated with local anesthetics.

A 17-gauge spinal cannula was introduced into the disc using the Kambin’s triangle approach. The Kambin’s triangle was formed by the path of the spinal nerve, upper border of the lower vertebral body, and anterior border of the superior articular process of the low vertebra.

After the cannula was positioned at the junction of the annulus and nucleus, the stylet was removed from the cannula, and the into-LB (intocare Co., Ltd., Yangju, Republic of Korea) was placed into the spinal cannula and advanced until its tip was approximately 5 mm beyond the tip of the spinal cannula. At this point, the active portion of the into-LB was situated beyond the inner layer of the annulus and within the nucleus. A total of 6 channels were created at the 2, 4, 6, 8, 10, and 12 o’clock positions. After the withdrawal of the into-LB, 2 mL of 0.3% mepivacaine was injected into the PPDD tract, but not intradiscally. The skin puncture site was then closed with a suture or Steri-Strips.

### 2.3. Epidurography

Before the PPDD, a wire-type epidural catheter, ABEL catheter (GS Medical, Cheongwon, Republic of Korea), was inserted via the sacral hiatus toward the anterior epidural space of the involved disc level under fluoroscopic guidance. After confirming the position of the wire-type epidural catheter, 5 cc of a nonionic contrast medium (Iopamiro 300 inj.; Bracco Imaging Korea, Ltd., Seoul, Republic of Korea) was injected (Figure 1a), and the physician assessed and recorded whether the contrast media spread above the involved disc level. In addition, the physician tried to insert the catheter upward past the involved disc level and recorded whether it was possible. After PPDD, a second epidurography was conducted to assess any changes in epidural spreading (Figure 1c,d). Then, the physician tried to insert the catheter more proximally again and recorded whether it was possible after PPDD, and finally the catheter was carefully removed.

### 2.4. Post-Procedure Care

All the patients were observed for 2–4 h postoperatively for any development of neurological deficit or other procedure-related complications. Before discharge, we provided patients with post-procedure precautions and information about rehabilitation treatment [16]. For the first 3 days, the patients were allowed to walk, stand, and sit for up to 10–20 min at a time, but the patients were instructed not to perform any lifting or bending activity during this period. No driving was allowed for the first 2 days. Return to sedentary or light work was permitted 3–4 days following PPDD. Lifting was limited to 3–4 kg during the first 2 weeks, and all the patients were prescribed 500 mg of oral cefadroxil bid for 5 days.

### 2.5. Evaluation of Outcome Variables

Each patient’s epidurograms (pre- and post-PPDD anteroposterior, lateral views) were analyzed by two pain physicians, who were not involved in the procedures and were only aware of the spinal level to be investigated. The analysis of the epidurograms was categorized as “improvement” or “no improvement.” Epidurographic improvement was defined as post-PPDD contrast media extending above the involved disc level in the anteroposterior view, although it did not extend above the involved disc level in pre-PPDD contrast media (Figure 1a,c). Epidurographic non-improvement was defined as no improvement in post-PPDD epidurograms. When the analyses differed, a third physician assessed the epidurograms, and a consensus was reached. After analysis of the fluoroscopic images, we divided the patients who underwent PPDD into two groups. The first, Group I, was defined as demonstrating an improvement in the epidurogram. The second, Group N, was defined as demonstrating no improvement in the epidurogram.

The patient’s degree of pain was measured using a standardized 11-point (0–10) NRS and evaluated by a well-trained physician at baseline 1 month after PPDD. The severity of pain was scored from 0 to 10, where “0” represented no pain and “10” represented the worst pain. The patients were encouraged to express their feelings regarding the pain.

The ODI was used to assess the patients’ degree of dysfunction. The ODI assessments were performed at baseline 1 month after PPDD. The ODI is a 10-item questionnaire used globally to functionally assess patients with low back pain. However, in this study, we used the 9-item Korean version of the ODI, which excludes the assessment of sexual function, for cultural reasons [17].

A successful treatment after 1 month was defined as over a 50% reduction in the NRS score post-PPDD. Further, we analyzed the relationship between epidurographic improvement and the success rate.

### 2.6. Statistical Analysis

When estimating the sample size by the pilot study, the total sample size was 28 patients when the significance level was 0.0500 and the power was 0.8 in the Chi-square test. However, since the number of patients is small and if w (Phi) is corrected, a suitable total sample size of 60 was obtained. The patients’ demographic data were analyzed using the Student’s t-test, Chi-square test, and Mann–Whitney U test. A Wilcoxon signed-rank test was used to determine the difference in NRS and ODI scores before and after PPDD. The relationship between epidurographic improvement and the success rate was analyzed using the Chi-square test.

## 3. Results

The demographic data are shown in Table 1. There was no significant difference between group I and group N. Changes in the NRS and ODI scores were analyzed before and 1 month after PPDD. Both NRS and ODI scores were significantly reduced in both groups. The NRS scores were significantly reduced between the groups, but there was no significant difference in ODI (Table 2). However, the success rate in group I was significantly higher than in group N (Table 3, *p* < 0.001).

## 4. Discussion

In this study, we found that PPDD is effective in reducing pain and improving functional capacity in patients who have chronic discogenic low back pain and herniated lumbar disc-related symptoms, regardless of the groups. Our results were similar to those of previous studies [7]. Although there were no significant differences in improvement of functional capacity between the two groups, the pain reduction and success rate were significantly higher in the epidurographic improvement group (Group I) than in the epidurographic non-improvement (Group N). Based on the results of this study, intra-procedure epidurography is a useful real-time assessment method for predicting successful PPDD outcomes, as well as a reliable determinant of the endpoint of the procedure.

PPDD mechanisms on the intervertebral discs have been well established [18]. PPDD shows its effect by down-regulating local inflammatory mediators, reducing disc size, and initiating the repair process. Ren et al. [19] found that PPDD effectively degraded phospholipase A2 (PLA2) activity. PLA2 activity is closely associated with intervertebral disc degeneration, intervertebral disc herniation, radicular pain, and lumbar discogenic pain. It is considered the rate-limiting enzyme in the inflammatory cascade reaction. They suggest that when intervertebral disc degeneration occurs, PLA2 is activated by various proinflammatory mediators, such as interleukin-1, tumor necrosis factor-α, and interleukin-6, which are secreted by the degenerative intervertebral disc. Interleukin-1, especially, is an important pathophysiologic factor in painful disc disorders [20].

The disc volume reduction effect of PPDD has been demonstrated in several studies [8,21]. Chen et al. [21] in a human cadaveric study showed that PPDD reduced intradiscal pressures significantly, especially in younger, healthy discs as compared to degenerative discs. Consequently, PPDD can alleviate nerve root compression and discogenic low back pain. Kasch et al. [22] evaluated this effect using 7.1 Tesla ultrahigh-field MRI in porcine discs. They showed volume reductions of 0.114 (SD: 0.054) mL, or 14.72% (thoracic) and 0.093 (SD: 0.081) mL, or 11.61% (thoracolumbar) compared with the placebo group.

Numerous studies have found favorable results with PPDD in the treatment of discogenic low back pain and herniated lumbar disc, especially contained disc protrusion [6,7,10,12,18,23,24,25,26,27,28,29,30,31]. The success rate of substantial pain relief post-PPDD varies from 6.3% to 84% [5,24,32,33]. Most studies have reported a success rate of >50%. Sharps and Isaac [11] showed the efficacy of PPDD. Overall 79% of 49 patients had a minimum of 2-points reduction on a VAS. Liliang et al. [9] reported that 21 patients (21/31, 67.7%) experienced substantial pain relief for an average period of 10 months (4–17 months). Furthermore, they reported that positive discography results prior to PPDD could improve and predict the success rate of the procedure, among other variables, such as age, sex, body mass index, hyper-intensity zone, Modic change, and spinal instability. In a systemic review and meta-analysis study, PPDD reduced VAS scores in the long term (24 months) and improved ODI scores. In addition, the review suggested that PPDD is a more effective, low-complication, and minimally invasive procedure compared to other treatments [7]. In two long-term follow-up studies conducted in China, PPDD was shown to be effective for pain relief and function improvement for 2–3 years. However, there is no significant difference between the 3- and 5-year postoperative VAS and ODI scores, and excellent or good patient satisfaction was 87.9% at 1 week, 72.4% at 1 year, 67.7% at 3 years, and 63.4% at 5 years [19,34].

The safety and efficacy of the PPDD procedure using Coblation^®^ technology have been analyzed. PPDD achieved a volumetric reduction of the disc tissues without overt thermal or structural damage to adjacent tissues [35]. Lee et al. [36] showed that PPDD does not rely on heat for tissue removal, and, therefore, does not introduce excessive heat that causes tissue damage in the disc. The temperature during PPDD is typically 40 °C to 70 °C, and the total reduction of thermal influence was 5 mm from the tip [35,37]. In a porcine model, the coblation channel had a clear coagulation boundary of the nucleus pulposus. In addition, there was no evidence of direct mechanical or thermal damage to the annulus and endplate, and neural elements of the spinal cord and nerve roots at the level of the procedure were observed in the histologic examination [38].

In addition to the pain and functional scores, there are various methods to verify the success of PPDD. Radiological imaging studies, including plain radiography, CT, and MRI have been used [8,22], and there are also studies that have evaluated the PPDD outcome by thermography [39,40]. The thermal difference and pain scores were improved after PPDD, but there was no significant correlation [40]. However, these methods do not provide detailed information and are expensive. Moreover, they are not available for real-time assessment in the procedure room; thus, a simple and quick intra-procedure morphological assessment of the adequacy of decompression is still required.

Traditionally, epidurography during epidural steroid injections provided safe and accurate therapeutic injection [41,42,43]. It is a simple, quick, real-time, less expensive, and relatively safe method. In addition, epidurography is used to determine the degree of epidural adhesion. Although advanced technology, including CT and MRI, have made significant advances in the diagnosis of epidural fibrosis, it is believed that epidural adhesions are best diagnosed by epidurography [44]. Notably, epidurography correlated with the success of the percutaneous epidural neuroplasty, severity of pain, relief of pain, and patient satisfaction [13,45]. Moreover, it is used to assess the adequacy of decompression during spinal surgery, which is very similar to this study [14].

In this study, we applied the advantages of epidurography to the decompression effect of a disc by PPDD. Before PPDD, we observed that the contrast media did not extend above the involved disc level. After PPDD, when the disc was decompressed, we observed that the contrast media extended above the involved disc level. Moreover, the wire-type epidural catheter, which had not been passed before, was passed in some cases. We assumed that the partial removal of the nucleus pulposus by radiofrequency energy ensured the epidural space. Similar to other studies’ results on the efficacy of epidurography mentioned above, our study showed that the high success rate of PPDD is correlated with an improvement in epidurography. Based on our results, intra-procedure epidurography may be a reliable method for ensuring adequate decompression of the disc. Therefore, it is expected that the patient’s outcomes can be improved by using intra-procedure epidurography in addition to the existing manufacturer guidelines during PPDD.

Our findings need to be interpreted within the limitations of the study. First, this study was retrospective and had a relatively small sample size. Second, we could not analyze the difference in disc volume change by MRI before and after PPDD because of the cost implications. Third, the different proportions of type and severity of disc herniation between the two groups, which may affect the success rate of the procedure, were not evaluated on MRI. Fourth, epidural adhesion was not evaluated. A herniated disc may cause inflammation within the epidural space, and it can also make epidural adhesion that affects contrast media spreading. Fifth, the observation periods were relatively short. A randomized, controlled, double-blind, long-term follow-up study should be conducted to support the finding of this study.

This study’s results showed that PPDD is an effective treatment method for patients with chronic discogenic low back pain and herniated lumbar disc-related symptoms. We also suggested that intra-procedure epidurography may be a potential PPDD outcome predictor for ensuring successful outcomes and determining the endpoint of the procedure.

## Figures and Tables

**Figure 1 jcm-11-07144-f001:**
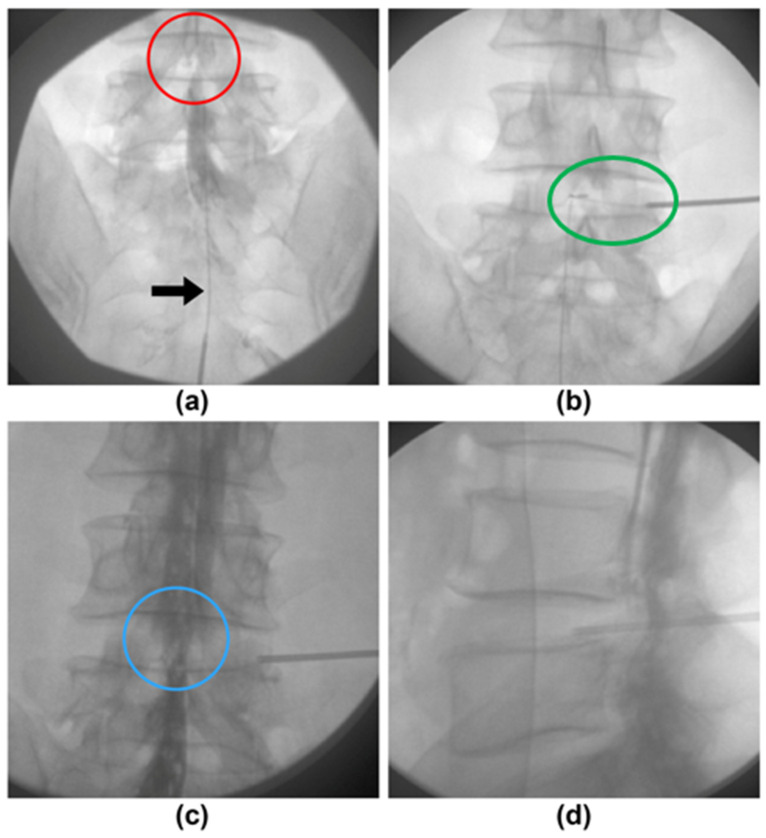
(**a**) Fluoroscopic anterior-posterior view of the pre-percutaneous plasma disc decompression (pre-PPDD). The filling defect was present at lumbar 4/5 disc level. (**b**) Fluoroscopic anterior-posterior view of the during-PPDD. (**c**) Fluoroscopic anterior-posterior view of the post-PPDD. The resolution of filling defect was verified. (**d**) Fluoroscopic lateral view of the post-PPDD. The 17-gauge spinal cannula was positioned at the junction of the annulus and nucleus Arrow, ABEL catheter; Red circle, the contrast media does not spread well above the involved disc level; Green circle, the into-LB was situated 6 o’clock position within the nucleus; Blue circle, the contrast media sufficiently spread above the involved disc level.

**Table 1 jcm-11-07144-t001:** Demographic data.

Variable	Improvement Group (*n* = 39)	Non-Improvement Group (*n* = 21)	*p*-Value
Sex			0.725 ^a^
Male	20 (51.28%)	9 (42.86%)	
Female	19 (48.72%)	12 (57.14%)	
Age, years			0.981 ^b^
Mean ± SD	58.51 ± 16.27	58.61 ± 15.60	
Height, cm			0.433 ^c^
Mean ± SD	164.08 ± 7.67	165.57 ± 6.82	
Weight, kg			0.053 ^b^
Mean ± SD	67.79 ± 11.29	62.27 ± 8.15	

^a^ Chi-squared test was used. ^b^ Student *t*-test was used. ^c^ Mann–Whitney test was used.

**Table 2 jcm-11-07144-t002:** Comparison of NRS score and ODI score before and after PPDD.

Variable	Before	After	Difference	^†^ *p*-Value
NRS				
Group I	7.56 ± 1.63	3.18 ± 1.76	4.38 ± 20.9	<0.001 ^a^
Group N	6.43 ± 1.16	4.71 ± 1.27	1.70 ± 1.19	<0.001 ^b^
^‡^ *p*-Value	0.005 ^d^	0.001 ^d^	<0.001 ^d^	
ODI				
Group I	39.44 ± 11.43	25.38 ± 11.43	14.05 ± 14.12	<0.001 ^b^
Group N	40.71 ± 12.10	32.62 ± 13.84	8.10 ± 7.66	<0.001 ^a^
^‡^ *p*-Value	0.692 ^d^	0.059 ^c^	0.368 ^d^	

^a^ Paired samples *t*-test was used. ^b^ Wilcoxon signed-rank test was used. ^c^ Independent samples *t*-test was used. ^d^ Wilcoxon rank sum test. ^†^ *p*-Value, difference in each groups. ^‡^ *p*-Value, difference between the two groups. PPDD, percutaneous plasma disc decompression; NRS, Numerical Rating Scale score; ODI, Oswestry Disability Index score; Group I, Improvement Group; Group N, Non-Improvement Group.

**Table 3 jcm-11-07144-t003:** Comparison of success rate between Group I and Group N.

	Group I	Group N	*p*-Value
Success rate	29/39 (74.36%)	3/21 (14.29%)	<0.001 ^a^

^a^ Chi-square test was used.

## Data Availability

The study’s data are available on request from the corresponding author.

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
