# Peer review of "A New Role for Epidurography: A Simple Method for Assessing the Adequacy of Decompression during Percutaneous Plasma Disc Decompression"

_jcm, 2022, doi:10.3390/jcm11237144_

Round 1

Reviewer 1 Report

Thank you for your academically and clinically valuable research.

Let me give you a few comments.

1. Materials and Methods

Please describe the two groups (improvement vs. non-improvement).

2. Results

Line 181) no significant difference between the groups (Table 2)

But in Table 2, total p-values are <0.001. 

Please clarify the description between the manuscript and the table.

3. References

Please number your references.

Author Response

Dear Editor

Thank you very much for reviewing this manuscript. This would have been very hard. We are grateful for your comments. And we have faithfully modified them. Please let us know if there is anything to modify again.

Above all things, we are very glad that you've recognized our attempts to improve the patient's outcome even a little.

Thank you for your consideration. We look forward to hearing from you.

Sincerely yours,

Reviewer 2 Report

This is an interesting paper with some interesting initial outcomes. The study is small so it will only indicate potential benefit but it shows good evidence for further investigation in this area.

There are some minor English errors but overall it is clear and well written.

Specifically I would like to see more in section 2.5 and Figure 1 to explain the analysis method better.

Also I would like to see a paragraph in the discussion to explain how this might be used clinically to improve patient outcomes.

Other specific comments are as follows:

* line 97 - could you give more details about the level of experience of the pain physican - did the same physician do all the procedures? Also figure 1 does not relate to the physician - please rephrase this sentence or split this up.

* Figure 1 Can you improve this diagram by marking more specific lines and annotation on the figure to indicate the method and points of interest to better explain the success and failure criteria of the analysis in section 2.5

* Section 2.5 - what was the criteria that the pain physicians were looking for? Can this be quantified? Is this just based on opinion? How many times did a third physician need to be used? If this is a lot then I don't think this technique seems very repeatable. You need to comment on this as a limitation. Just the fact that a 3rd physician is needed means that the method of analysis is very subjective rather than scientific.

Please can you explain this more? And/or can you be more specific on the criteria for these decisions - marking lines on figure 1 would help a lot with this I think

* Line 171 - please define N and w or use words

* Table 2 please define group N and group I - you need to explain all symbols in full

* Line 200 - I think you need to explain that I and N relate to the analysis of the epidurography assessment - please make this clearer for the reader. This needs to be expanded and carefully worded

Also need to define I and N groups better

* Line 201 - can you quantify the level of improvement - what percentage difference would this make for example?

* Line 288 (discussion section in general) - I am still unclear about where in the care pathway this would be used could you talk about this in the discussion. E.g. How using this method could change patient outcomes and how you recommend it be used if you had better evidence from a larger study showing its efficacy . 

Author Response

(The authors gave the same response as above.)
